# TOTUM-854 Human Circulating Bioactives Preserve Endothelial Cell Function

**DOI:** 10.3390/nu17081331

**Published:** 2025-04-11

**Authors:** Fabien Wauquier, Doriane Ripoche, Line Boutin-Wittrant, Yolanda F. Otero, Stéphanie Krisa, Josep Valls, Mahéva Maura, Florian Le Joubioux, Thierry Maugard, Gaëtan Bolea, Grégory Meyer, Cyril Reboul, Véronique Roux, Nicolas Macian, Gisèle Pickering, Bruno Pereira, Maxime Bargetto, Véronique Sapone, Murielle Cazaubiel, Sébastien Peltier, Pascal Sirvent, Yohann Wittrant

**Affiliations:** 1Clinic’n’Cell SAS, Faculté de Médecine, 28 Place Henri Dunant, 63001 Clermont-Ferrand, France; fabien.wauquier@clinicncell.com (F.W.); line.wittrant@clinicncell.com (L.B.-W.); 2Valbiotis, 20 rue Henri et Gilberte Goudier, 63200 Riom, France; ripochedoriane@gmail.com (D.R.); pascal.sirvent@valbiotis.com (P.S.); 3INRAE, INP, Campus ISVV, Université de Bordeaux-INRAE-INP-ISVV, 210 Chem. de Leysotte, 33140 Villenave-d’Ornon, France; stephanie.krisa@u-bordeaux.fr; 4MetaboHUB, Bordeaux Metabolome, 210 Chem. de Leysotte, 33140 Villenave-d’Ornon, France; josep.valls-fonayet@u-bordeaux.fr; 5Valbiotis, Zone Industrielle des 4 Chevaliers, Bâtiment 12F, Rue Paul Vatine, 17180 Perigny, France; maheva.maura@valbiotis.com (M.M.); florian.lejoubioux@valbiotis.com (F.L.J.); bargetto.maxime@gmail.com (M.B.); veronique.sapone@valbiotis.com (V.S.); murielle.cazaubiel@valbiotis.com (M.C.); sebastien.peltier@valbiotis.com (S.P.); 6CNRS, LIENs, Campus NA, La Rochelle Université CNRS-LIENSs-, 2, Rue Olympe de Gouges, 17000 La Rochelle, France; thierry.maugard@univ-lr.fr; 7LAPEC EA-4278, Avignon Université, 228 Route de L’Aérodrome, 84000 Avignon, France; g.bolea@biophysium.com (G.B.); gregory.meyer@univ-avignon.fr (G.M.); cyril.reboul@univ-avignon.fr (C.R.); 8CIC INSERM 1405/Plateforme d’Investigation Clinique CHU Gabriel Montpied, 63000 Clermont-Ferrand, France; v_morel@chu-clermontferrand.fr (V.R.); nmacian@chu-clermontferrand.fr (N.M.); gisele.pickering@uca.fr (G.P.); 9Biostat Unit, DRCI, CHU, 63000 Clermont-Ferrand, France; bpereira@chu-clermontferrand.fr; 10INRAE, UMR 1019, UNH, 63000 Clermont-Ferrand, France

**Keywords:** clinical trial, ex vivo, hypertension, plant extract, oxidative stress, lipotoxic stress, human metabolites, human endothelial cells

## Abstract

**Background**: TOTUM-854 is a patented plant extract blend characterized by its components that have previously been described for their potential health benefits in limiting hypertension onset. However, most of the literature data remain descriptive regarding the mode of action at the cellular level, especially in humans, and further investigations are required for optimized therapeutic strategies. **Methods**: We first demonstrated in an L-NAME mouse model that TOTUM-854 supports the prevention of hypertension in vitro and in vivo. Then, we designed an ex vivo clinical innovative approach considering the circulating metabolites produced by the digestive tract upon TOTUM-854 ingestion in humans. Human serum was collected in healthy volunteers before and after the acute intake of 3.71 g of TOTUM-854. The bioavailability of circulating metabolites was confirmed and characterized by UPLC-MS. Human serum containing TOTUM-854-derived metabolites was further processed for incubation with human endothelial cells (HUVECs), in the absence or presence of palmitate (200 µM). **Results**: HUVEC protection against lipotoxicity was characterized by (1) decreased ACE-1 activity (−32% *p* < 0.0001); (2) the inhibition of oxidative stress with decreased ROS (−12% observed by DCFDA and DHE fluorescent microscopy) and decreased Nox2 gene expression (−6.7 fold change vs. palmitate, *p* < 0.01); and (3) the inhibition of an inflammatory response, with a decrease in IL-1β release (−37% compared to palmitate, *p* < 0.001) and decreased MCP-1 and VCAM-1 gene expression (−93% *p* < 0.001 and −77% *p* < 0.001, respectively). **Conclusions**: Overall, this study provides insightful data regarding the protective role of TOTUM-854 in human endothelial cells. Using an innovative clinical ex vivo approach, our data support the role of TOTUM-854 circulating metabolites in vascular protection in humans.

## 1. Introduction

Blood pressure is highly correlated with both cardiovascular morbidity and mortality [1]. To date, hypertension affects over 1.2 billion people worldwide and critically impacts public health systems [2]. Hypertension is currently defined as systolic or diastolic blood pressure above 140 mmHg and 90 mmHg, respectively, alone or in combination [3]. Ninety percent of diagnosed patients suffer from essential hypertension, the etiology of which remains to be determined. Hypertension is a multifactorial disease involving environmental, genetic, and behavioral factors [4]. Family-based studies have shown that genetic factors contributing to the variability of either systolic and/or diastolic blood pressure represent 30 to 60% [5]. Risk-conferring behaviors such as smoking, drugs, a lack of physical activity, obesity, and diet also potently contribute to hypertension onset [6].

Hypertension relies on an angiotensin-related mechanism that is widely supported by a large amount of experimental, clinical, and genetic evidence. Indeed, blood pressure is tightly regulated by the renin angiotensin aldosterone system (RAAS). Among the key protagonists, ACE-1 (angiotensin converting enzyme-1) converts angiotensin I into angiotensin II (AngII) that binds its cognate receptors (AGT1R and AGT2R) on target tissues, including the brain (thirsty and vasopressin), the kidneys (aldosterone and reabsorption), and vessels (arteriolar vasoconstriction), leading to the increase in both the extracellular water volume and the total peripheral resistance [7,8].

Along with this “angiotensin-focused” point of view, a few original studies have proposed that the skin, the immune system, and the peripheral vascular resistance could fulfill the global picture, all of them sharing inflammation, immune cell infiltration, ROS activation, or loss of vascular capabilities as a common starting point [4,9]. Obesity and diet are risk factors that potently contribute to endothelial cell dysfunction and further hypertension onset [6]. To date, hypercholesterolemia and hypertriglyceridemia have been shown to drive atherosclerosis onset, promoting local inflammation and oxidative stress that finally alter endothelial cells’ abilities to regulate vessel tone and peripheral resistance [10,11].

In this context, lifestyle interventions such as dietary manipulation, including salt or calorie restrictions, represent the initial step in the treatment of hypertensive patients, regardless of the grade of HP [12]. In cases where BP remains above 150/95 mmHg, these lifestyle interventions may be complemented with immediate drug treatment such as diuretics and/or angiotensin-converting enzyme (ACE) inhibitors. However, the side effects reported for both and ACE inhibitors are widely associated with dry cough (mostly), hypotension, hyperkalemia, and angioedema [13]. Thus, dietary interventions remain the first-line strategy, and nutritional strategies have become a major field for therapeutic innovation.

Nutrition is our first source of bioactive compounds and therefore may play a key role in maintaining or preserving body homeostasis. Among those bioactives, polyphenols, which are phenolic molecules belonging to the plant immune system, have widely recognized antioxidant and anti-inflammatory properties [14]. Their health benefits have particularly been reported in the cardiovascular system [15]. Regarding their role in chronic vascular diseases, in the last five years, nearly one-third of the total articles related to this theme demonstrated the dynamics of nutritional approaches in the field.

To date, ex vivo studies have evidenced that the phenolic compounds present in fruit and vegetable juices could induce potent vasorelaxation in aorta segments in rats [16]. Earlier, in a preclinical model of L-NAME-induced hypertension, garlic consumption completely prevented the increase in systolic blood pressure [17]. Bilberry extract also demonstrated an inhibitory effect on the establishment of induced pulmonary arterial hypertension in Wistar rats [18]. In a spontaneous model of hypertension in rats, the consumption of chlorogenic acid reduced hypertension and led to an improvement in endothelial function [19]. Likewise, in a model of cyclosporine-induced hypertension, both chlorogenic acid and caffeic acids limited angiotensin-converting enzyme activity and oxidative stress [20,21]. However, clinical evidence on how it may work in humans at the cellular level is lacking. 

TOTUM-854 is a polyphenol-rich blend composed of six different plant extracts. Each compound has been previously shown, separately, to impact inflammation and oxidative stress. The potential synergistic properties still need to be deciphered. The aim of this study was to demonstrate if and how this polyphenol combination can maintain, improve, or preserve endothelial function in order to support preventive strategies in the mid to moderately hypertensive population.

The potential health value of TOTUM-854 was first evidenced both in vitro and in a preclinical model of L-NAME-induced hypertension. Then, according to our ex vivo clinical approach [11,22,23,24,25,26,27,28], we collected human bioactive metabolites in the bloodstream resulting from TOTUM-854 ingestion and examined their influence on human endothelial cells’ behavior and functions in the context of a lipotoxic environment.

## 2. Materials and Methods

### 2.1. Study Product

VALBIOTIS has formulated TOTUM-854 (Patent: FR1460064), a food supplement resulting from the combination of six plant extracts (chrysanthellum indicum, artichoke leaves (*Cynara scolymus* L.), olive leaves (*Olea europaea* L.), blueberry (*Vaccinium myrtillus* L.), *Piper nigrum* L., and garlic (*Allium sativum* L.)). The blend was formulated according to previously described clinical data and the traditional uses for their supposed benefits in preventing hypertensive conditions. The formulation of the product meets the requirements of the European regulations relating to food supplements (European directive n°2002/46/EC and French decree n°2006-352 of 20 March 2006, which transposes it).

### 2.2. Ethics Preclinical Study

All animal experiments were performed according to the European Parliament Directive 2010/63/EU and were approved by the local research ethics committee (C.E. Languedoc-Roussillon, France, under the number CEEA-00322.03).

### 2.3. Ethics Clinical Trial

This study was conducted in accordance with the Declaration of Helsinki of 1975 (https://www.wma.net/what-we-do/medical-ethics/declaration-of-helsinki (accessed on 12 November 2021)) revised in 2013. The human study was approved by the French Ethical Committee (N° SI RIPH: 21.03989.000023/N° ID RCB: 2021-A02695-36/Comité de Protection des Personnes CPP Est III; approved 8 December 2021). Before they participated in the project, all volunteers were informed of the objectives and the potential risks of the present study and provided their written informed consent.

### 2.4. In Vitro ACE Inhibition Assays

*Reagents:* angiotensin-converting enzyme from rabbit lung (ACE), Hippuryl-His-Leu acetate salt, Hippuric acid, sodium hydroxide, sodium chloride, potassium chloride, boric acid, trifluoroacetic acid, captopril, and dimethyl sulfoxide (DMSO) were obtained from Merck/Sigma-Aldrich (Darmstadt, Germany). Acetonitrile was obtained from Carlo-Erba (Val-de-Reuil, France). Pure water was obtained using an Evoqua ultra-pure water production system (Water Technologies, Günzburg, Germany). *ACE inhibition assay:* ACE hydrolytic activity was assessed by monitoring the conversion of Hippuryl-His-Leu (HHL) into Hippuric acid (HA) in a borate buffer (0.05 M, pH 8.3) using an HPLC-UV. A commercial reference inhibitor, captopril, was used as a positive control.

The relative ACE activity was calculated as follows: ACE activity (%) = 100 × (A2 − A0)/(A1/A0) where A0 is the Hippuric acid peak area in Blank, A1 is the Hippuric acid peak area in the negative control, and A2 is the Hippuric acid peak area in the sample with the potential inhibitor.

### 2.5. Preclinical Protocol

Twelve-week-old C57Bl6/J mice were divided into three groups: a control group, drinking tap water (n = 11); an L-NAME group, receiving L-NAME-supplemented drinking water (100 mg/kg/day) (n = 11); and an L-NAME + TOTUM-854 group, receiving L-NAME-supplemented drinking water (100 mg/kg/day) and TOTUM-854 administration (500 mg/kg/day) via daily oral gavage (n = 11). The control and L-NAME groups received the vehicle via daily oral gavage. Body weights were recorded weekly at the same time as bedding changes. Arterial blood pressure was measured using the CODA Tail-Cuff System (Kent Scientific Corporation, Torrington, CT, USA) after one week of acclimation, at the beginning (T0), and once a week for 3 weeks. All measurements were performed by the same trained operator and in the same time frame (from 9 a.m. to 1 p.m.). All blood pressure data were obtained by taking the average of at least 7 measurements (from 15 measuring cycles per machine).

### 2.6. Human Study Design and Pharmacokinetic of Absorption

A total of 10 healthy volunteers (age: 23.7 years old, +/−3.97; BMI: 22.29 kg/m^2^, +/−1.18; >60 kg; absence of pathologies or drug treatment was required; and no distinction in ethnicity) were enrolled in this study. They were monitored for normal blood formulation and renal (urea and creatinine) and liver functions (including aspartate aminotransferase (AST), alanine aminotransferase (ALT), and gamma-glutamyltransferase (GGT) activity).

The first phase of the clinical approach characterized TOTUM-854’s metabolite absorption profile. Volunteers fasted for at least 12 h prior to the ingestion of 3710 mg of TOTUM-854. TOTUM-854 was provided as eight capsules. Serum samples (from venous blood) were collected before ingestion and every 20 min for 240 min after the ingestion of TOTUM-854. Samples were then aliquoted and stored at −80 °C until analyses. Blood sample collection was conducted by the Clinical Investigation Center—Inserm 1405, University Hospital of Clermont-Ferrand (ethical clinical certification NF S 96900).

Human circulating polyphenolic bioactives were characterized by ultra-high-performance liquid chromatography combined with tandem mass spectrometry (UPLC-MS/MS).

In the second phase of the clinical approach, volunteers were recalled to the clinical center for the collection of the metabolites’ enriched serum fractions. They were asked to fast for 12 h prior to being given 3710 mg of TOTUM-854. The collection consisted of two serum samples. One sample was collected before ingestion as the naïve serum fraction (baseline) and the second sample was collected after ingestion at Cmax (absorption profile’s peak) as the enriched serum fraction. Serum aliquots were stored at −80 °C until analyses.

### 2.7. Phenolic Compounds Extraction from Serum

The serum was mixed with 100% methanol for 1 min (1:3 *v*/*v*). The mixture was then centrifuged at 20,000× *g* for 15 min. The supernatant was then evaporated and dried. The dried material was dissolved in methanol/water (50:50 *v*/*v*). After agitation (1 min), ultrasonication (1 min), and centrifugation (20,000× *g* for 15 min), the supernatant was stored at −20 °C until analysis by ultra-high-performance liquid chromatography combined with tandem mass spectrometry.

### 2.8. Ultra-High-Performance Liquid Chromatography–Mass Spectrometry (UPLC-MS/MS)

Analyses were carried out using a 1260 Infinity UHPLC (Agilent Technologies, Santa Clara, CA, USA) system coupled to a 6430 triple quadrupole mass spectrometer from Agilent Technologies. Solvent A (water/formic acid 99.9:0.1, *v*/*v*) and solvent B (acetonitrile/formic acid 99.9:0.1, *v*/*v*) were used as a mobile phase at a flow rate of 0.3 mL/min, and the gradient in solvent A was as follows: 0 min 1% B, 2 min 5% B, 3 min 25% B, 6 min 25% B, 8 min 40% B, 11.5 min 95% B, 14 min 95% B, and 16 min 1% B. The MS/MS parameters were as follows: negative ion mode, capillary tension of 3000 V, nebulizer at 15 psi, dry gas of 11 L/min, a dry temperature of 350 °C, and acquisition in multiple reaction monitoring (MRM); see Appendix A for further details. The data were processed using MassHunter software (V.12) Agilent Technologies, Santa Clara, CA, USA).

### 2.9. Human Umbilical Vein Endothelial Cell Cultures

The primary human umbilical vein endothelial cells (HUVECs) were obtained from Thermo Fisher Scientific (C0155C, Villebon sur Yvette, France). During the maintenance phase, HUVECs were cultured in Complete EGM^TM^-2 Endothelial Cell Growth Medium-2 (CC-3162, Lonza, Levallois-Perret, France). All cell cultures were performed at 37 °C in an atmosphere of 5% CO2/95% air. To analyze the effects of TOTUM-854 metabolites, the cells were preincubated for 24 h in basal EGM^TM^-2 in the presence of 10% of human serum (naïve or containing circulating metabolites) according to the Clinic’n’Cell methodology (DIRV INRAE 18-0058), prior to an additional 24 h incubation in a palmitate-induced lipidic stress environment (palmitate 200 µM).

### 2.10. Preparation of Palmitate Solution

Palmitate (Sigma, Saint-Quentin-Fallavier, France) dissolution proceeded as follows: incubation with bovine serum albumin (BSA; Saint-Quentin-Fallavier, France, Sigma) and dissolution in pure ethanol at 70 °C. The final concentration was set to 500 mmol/L. A pre-warmed BSA solution (10% *w*/*w*, 37 °C) was mixed with this stock solution to reach a final concentration of 5 mmol/L. Mixture clarification occurred via two sessions of incubation at 55 °C for 15 min. The final molar ratio was set at 3.2:1 (palmitate–BSA). The control vehicle proceeded in the same conditions. 

### 2.11. Cell Viability

An XTT-based method was used to determine the cell viability (Cell Proliferation Kit II, Sigma-Aldrich, Saint-Quentin-Fallavier, France). The experimental procedures followed the supplier’s recommendations. Measurements were performed in hexaplicate for each sample condition of the ten volunteers. OD was measured at 450 nm.

### 2.12. Angiotensin-Converting Enzyme 1 Assays

ACE-1 activity was tested in endothelial cells using ACE1 assay kits from Abcam (ab273308, Abcam, Paris, France). The experimental procedures were set according to the supplier’s recommendations. The kit utilizes the ability of ACE1 to hydrolyze a synthetic substrate, which results in a decrease in optical density at 345 nm. Measurements were performed in quadruplicates for each sample condition of the ten volunteers.

### 2.13. Dihydroethidium (DHE) Staining

The DHE (Dihydroethidium) Assay Kit (ab236206, Abcam, Paris, France) was used to measure reactive oxygen species (ROS) directly in live cells. This kit uses DHE as a fluorescent probe for the detection of ROS generation, notably superoxide, hydrogen peroxide, and non-specific oxidation by other sources of ROS to form ethidium (excitation at 480–520 nm and emission at 570–600 nm) [29]. At the end of the incubation period, with palmitate in the presence of naïve or enriched serum, cells were incubated with 5 µM of the DHE probe for 30 min at 37 °C and washed prior to observation. Live cells were imaged using a ZEISS optical microscope (ZEISS, Rueil Malmaison, France) with a B/W cooled camera and analyzed using ImageJ software (NIH) (version 1.54f). Measurements were performed for each sample condition of the ten volunteers.

### 2.14. Measurement of Nitric Oxide (NO)

A Nitrate/Nitrite (final products of NO) colorimetric assay kit was obtained from Cayman Chemical (references 780001, Cayman Chemical, Ann Arbor, MI, USA). Assays were conducted according to the manufacturer’s instructions. The levels of NO were examined in culture supernatants. Measurements were performed in quadruplicate for each sample of the ten volunteers.

### 2.15. IL-1β Release

IL-1β levels were evaluated in the cell culture supernatant using Human IL-1 beta ELISA Kit (abcam—ab214025, Paris, France) according to the manufacturer’s recommendations. Measurements were performed in quadruplicate for each sample condition of the ten volunteers.

### 2.16. Real-Time RT-qPCR

mRNA from HUVECs was isolated using TRIzol™ Reagent (Ambion—Life Technologies) according to the supplier’s recommendations. VCAM-1 (vascular cell adhesion molecule 1), MCP-1/CCL2 (monocyte chemoattractant protein 1), and NOX2 (NADPH oxidase 2) mRNA expression levels were measured by RT-qPCR (PowerUp SYBRgreen, Applied Biosystems). β-Actine was used as a housekeeping gene. Primers were designed as follows: VCAM-1-F: 3′- ACT TGA CTG TGA TCG GCT T -5′; VCAM-1-R: 3′- TGA CTC CGT CTC ATT GAC TTG -5′; MCP-1-F: 3′- GCC TCT GCA CTG AGA TCT TC -5′; MCP-1-R: 3′- AGC AGC CAC CTT CAT TCC -5′; NOX2-F: 3′- ATT CCT GTC CAG TTG TCT TCG -5′; NOX2-R: 3′- CTG ATT CTC TTG CCA GTC TGT’; ACTβ-F: 3′-ATT GGC AAT GAG CGG TTC-5′; ACTβ-R: 3′-GGA TGC CAC AGG ACT CCA-5′.

### 2.17. Statistics

Prism V.7.0, 8.0, and V.9.4.1 (GraphPad Software) were used to run statistical tests and draw figures. The following statistical plan was applied: a Shapiro–Wilk normality test was used to determine whether the data were consistent with a Gaussian distribution. If the data were not distributed according to a normal distribution, a Kruskal–Wallis non-parametric test was used, followed by the Dunn test for post hoc comparison. When a normal distribution and equal variance were assumed, the measures were subjected to a one-way ANOVA followed by Tukey’s test for multiple comparisons. In the case of measurements repeated over time, the differences between groups and time points were tested using a repeated-measures two-way ANOVA followed by Sidak’s post hoc test for multiple comparisons. If a piece of data was missing, making it impossible to run a repeated-measures two-way ANOVA, mixed-effect analysis was used instead. In Figure 1 and Figure 2, the values are expressed as mean ± SEM. In Figure 3, data are expressed as mean ± SD. In Figure 4, Figure 5, Figure 6 and Figure 7, values are presented by boxes that indicate the median and interquartile range (lower and upper) and by whiskers that indicate the minimum and maximum. Differences were considered statistically significant at *p* < 0.05, with * representing *p* < 0.05, ** representing *p* < 0.01, *** representing *p* < 0.001, and **** representing *p* < 0.0001. Finally, ns represents *p* > 0.05.

## 3. Results

### 3.1. In Vitro and In Vivo First Evidences of the TOTUM-854 Potential in Regulating Blood Pressure

To test the potential benefit of TOTUM-854 on blood pressure, we first questioned its capacity to inhibit ACE activity in vitro. As shown in Figure 1, TOTUM-854 dose-dependently decreased ACE activity. This inhibition nicely paralleled the effect of Captopril, a synthetic analog of the ACE-inhibiting peptide found in the Jararaca snake venom and a drug of reference that is widely used to treat hypertension. TOTUM-854’s IC50 was about 9.869 g/L, while Captopril had an IC50 of 10.31 nM (2.2 µg/L). Thus, the pharmacological capacity of Captopril to inhibit ACE activity was found to be much higher than that of TOTUM-854 (about 5.10^6^ times); still, these data were very encouraging from a nutritional point of view.

**Figure 1 nutrients-17-01331-f001:**
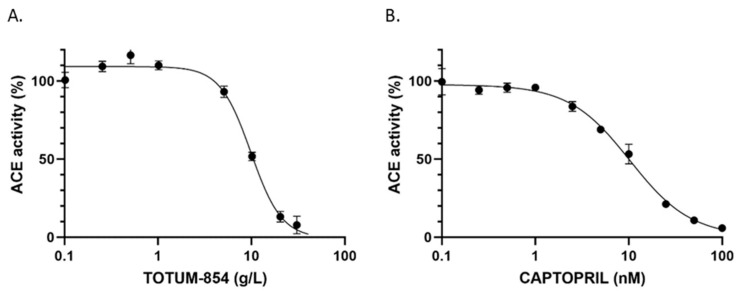
In vitro ACE inhibition assays by TOTUM-854 (**A**) and Captopril (**B**).

A commercial reference inhibitor, captopril, was used as a positive control. The IC50 of each compound corresponds to the lowest concentration at which ACE activity was halved (n = 2–6). Data are expressed as mean ± SEM.

To further investigate the relevance of such ACE inhibition in a more physiological model, TOTUM-854 was subjected to in vivo investigations (Figure 2A–D). L-NAME is a convenient analog to inhibit NO synthetase and block the production of NO, thus impairing endothelial function and driving hypertension onset. In our preclinical model, mice receiving L-NAME consistently showed an increase in blood pressure. Blood pressure increased over time during the 3 weeks in the L-NAME group (compared to the group’s baseline at day 0) to reach 27.8% and 24.9% for diastolic and systolic blood pressure, respectively. In contrast, when mice were given L-NAME in combination with TOTUM-854 supplementation, the rise in blood pressure at 3 weeks was constrained and limited to 7.9% and 8.7% for diastolic and systolic blood pressure, respectively (compared to the group’s baseline at day 0). The variation in blood pressure in the control group was up to 4.3% and 0.9% for diastolic and systolic blood pressure, respectively (compared to the group’s baseline at day 0), and there was no significant difference compared to the L-NAME+TOTUM-854 group.

**Figure 2 nutrients-17-01331-f002:**
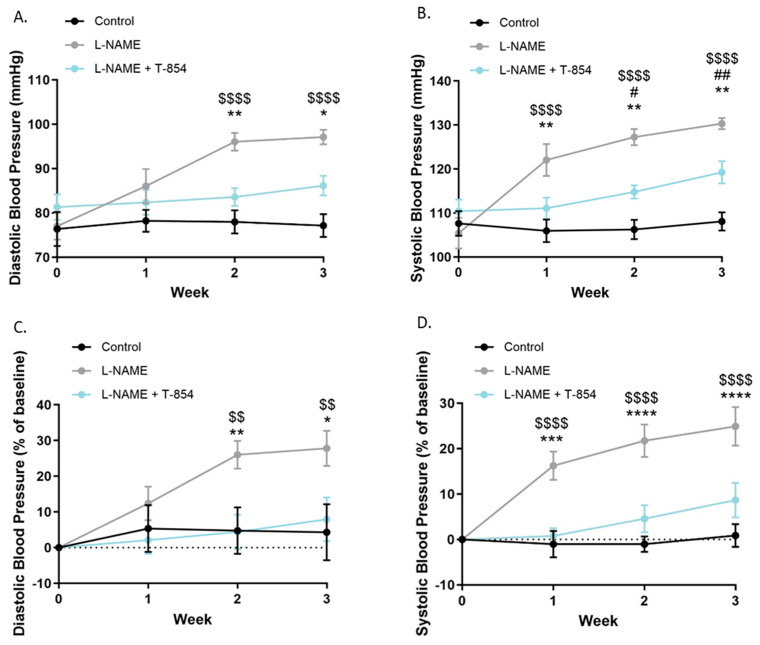
In vivo investigation of TOTUM-854 influence on blood pressure in mice. L-NAME was used as a convenient analog to blunt NO release and drive hypertension onset. Panels represent raw blood pressure values ((**A**): diastolic blood pressure; (**B**): systolic blood pressure) and blood pressure percentage of baseline ((**C**): diastolic blood pressure; (**D**): systolic blood pressure) over three weeks of L-NAME and T-854 administration. Data are expressed as mean ± SEM. L-NAME vs. L-NAME + T-854: * *p* < 0.05; ** *p* < 0.01; *** *p* < 0.001; **** *p* < 0.0001. Control vs. L-NAME + T-854: # *p* < 0.05; ## *p* < 0.01; L-NAME vs. Control: $$ *p* < 0.01; $$$$ *p* < 0.0001 (n = 11).

### 3.2. Kinetic of Apparition of Circulating Metabolites Resulting from TOTUM-854 Ingestion in Humans

TOTUM-854 was ingested and digested by fasted volunteers, and the occurrence of circulating metabolites in the blood was monitored using a kinetic approach to determine the time frame of the absorption peak. We found nine detectable circulating human metabolites, including two oleuropein glucuronides isomers, three luteolin glucuronides isomers, one hydroxytyrosol sulfate isomer, one ferulic acid sulfate isomer, one homovanillic acid sulfate isomer, and one dihydrocaffeic acid sulfate isomer (Figure 3). The chromatograms of the detected metabolites in serum are presented in the Appendix A. These data evidence the efficient absorption and bioavailability of the product. The Tmax (time to reach the maximum concentration observed in serum) ranged from 40 min to 180 min post-absorption, depending on the type of metabolites. According to the global absorption curves, and in order to obtain the most diverse enrichment of serum in TOTUM-854 metabolites, blood was collected before ingestion (naïve fraction) and at 60 min post-ingestion (enriched fraction) for the ex vivo investigation of HUVECs.

**Figure 3 nutrients-17-01331-f003:**
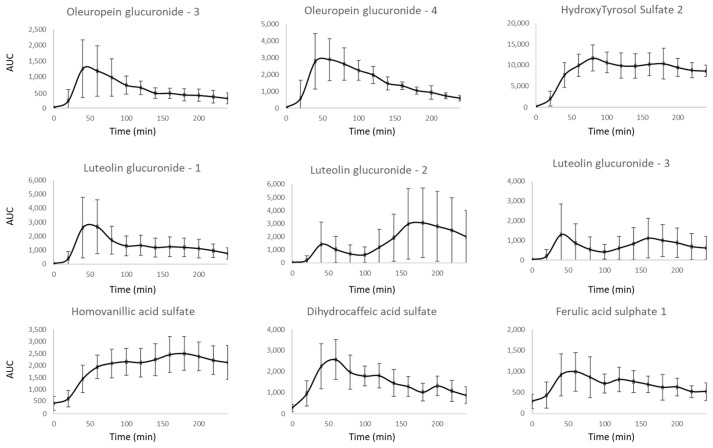
Metabolomic profiles in human serum following TOTUM-854 ingestion. Circulating metabolites resulting from TOTUM-854 ingestion were determined using ultra-high-performance liquid chromatography–mass spectrometry (UPLC-MS/MS) (AUC: Area under curve). Humans n = 10 volunteers.

### 3.3. Absence of Impact on Cell Culture Growth Following Incubation of HUVECs with Either Naïve or Metabolite-Enriched Human Serum

To avoid any bias in our conclusions, we first ensured that human serum (either naïve or containing circulating metabolites resulting from TOTUM-854 ingestion) had no detrimental effect on HUVEC growth and viability (Figure 4). HUVECs were pre-incubated for 24 h in the presence of either naive serum or TOTUM-854-enriched serum 24 h prior to lipotoxic stress induction by palmitate (200 μM). The data showed no differences between the different serum conditions. Neither naïve nor enriched human serum modified cell viability compared to each other or compared to the control FCS treatment. In contrast, the presence of palmitate slightly but significantly reduced cell growth. Interestingly, palmitate had no effect on cells when HUVECs were pre-incubated with the circulating metabolites from TOTUM-854.

**Figure 4 nutrients-17-01331-f004:**
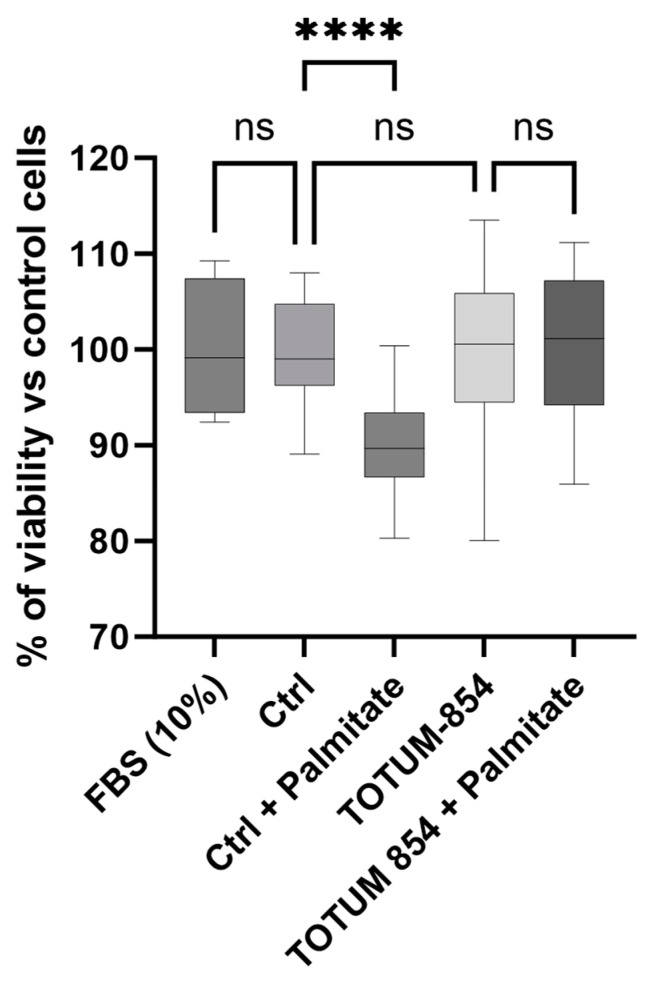
Cell viability (human umbilical vascular endothelial cells (HUVECs)). Cell viability was determined using an XTT-based method. Fetal Bovine Serum (FBS) was used as the reference (100% viability). Control groups (Ctrl) consist of human naïve serum exposition with or without palmitate. TOTUM-854 groups consist of human serum containing circulating metabolites from TOTUM-854 ingestion with or without palmitate. Measurements were realized in hexaplicate for each volunteer (n = 10 volunteers). Boxes indicate median and interquartile range (lower and upper), while whiskers indicate minimum and maximum ****: *p* < 0.0001; ns: *p* > 0.05.

### 3.4. TOTUM-854 Human Metabolites Decrease Angiotensin-Converting Enzyme 1 Activity Under Palmitate Stress

Angiotensin-converting enzyme 1 is a key player in the regulation of endothelial function. In this model, the presence of lipotoxic stress induced by the presence of palmitate consistently tends to increase ACE-1 activity without reaching significance. Metabolites from TOTUM-854 had no impact on ACE-1 activity on their own, but interestingly, in the presence of palmitate, they significantly decreased ACE-1 activity (−31% *p* < 0.0001) in HUVECs, decreasing ACE-1 activity to baseline in this lipotoxic stress environment (Figure 5).

**Figure 5 nutrients-17-01331-f005:**
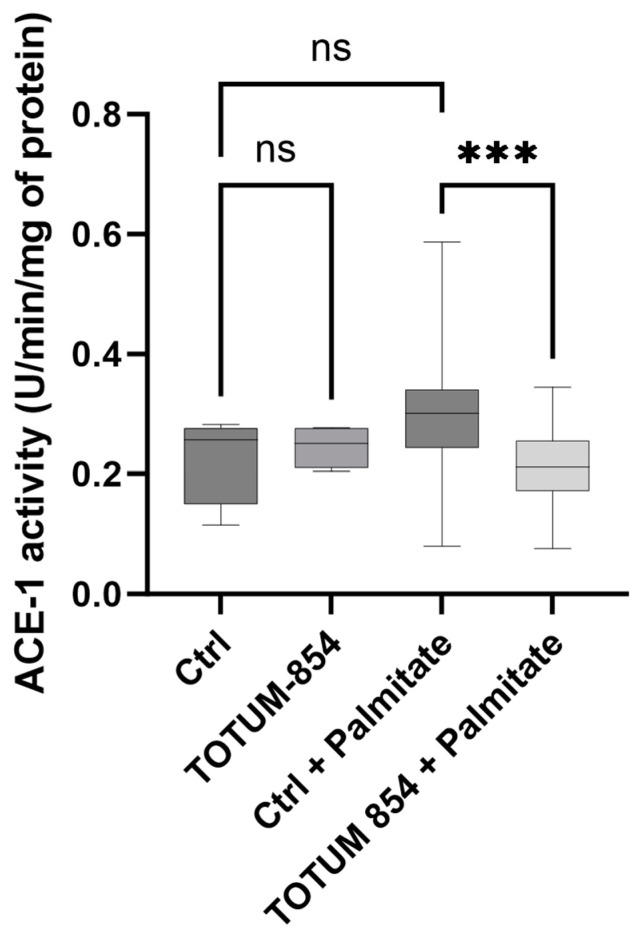
Influence of TOTUM-854 human metabolites on ACE-1 activity in human umbilical vascular endothelial cells (HUVECs). Control groups (Ctrl) consist of human naïve serum exposition with or without palmitate. TOTUM-854 groups consist of human serum containing circulating metabolites from TOTUM-854 ingestion with or without palmitate. Measurements were realized in triplicate for each volunteer (n = 10 volunteers). Boxes indicate median and interquartile range (lower and upper), while whiskers indicate minimum and maximum ***: *p* < 0.001; ns: *p* > 0.05.

### 3.5. TOTUM-854 Metabolites Limit Oxidative Stress in a Lipotoxic Environment

Oxidative stress, and more specifically the production of reactive oxygen species, is a driving force of endothelial tissue dysfunction. Using a DHE probe, we investigated the production of ROS in our different conditions (Figure 6A,B). Metabolites from TOTUM-854 had no impact on their own in the absence of palmitate. As expected, DHE staining increased in the presence of palmitate, evidencing a consistent increase in ROS production in a lipotoxic context (+130%). However, the presence of TOTUM-854 metabolites limited the increase in DHE staining (−25% compared to palmitate alone or −46% for the combination). In other words, cells were protected from this palmitate-induced oxidative stress when they were pre-incubated with TOTUM-854 metabolites (DHE’s increase was almost halved). NADPH oxidases, and most notably NOX2, are responsible for ROS production in endothelial tissues. Interestingly, NOX2 expression nicely correlates with DHE staining. Although the decrease in NOX2 expression by TOTUM-854 metabolites alone did not reach significance, in the presence of palmitate, they significantly reduced its expression by 0.58 log^10^ (−3.8 fold) (Figure 6C). We also investigated NO release as a master endothelium hypotensive mechanism. Neither palmitate nor TOTUM-854 metabolites had any impact on NO production by HUVECs (Figure 6D).

**Figure 6 nutrients-17-01331-f006:**
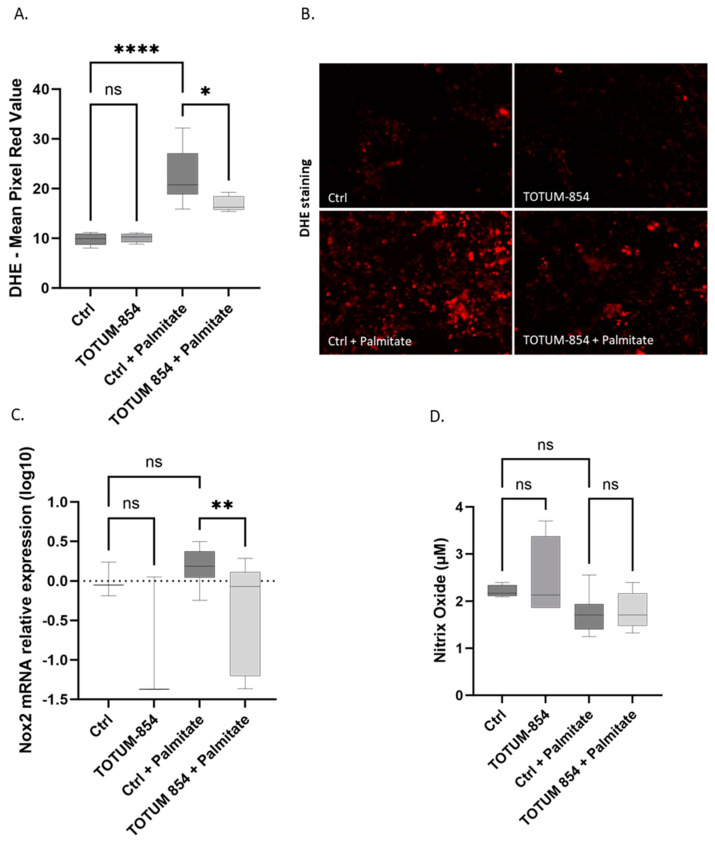
Influence of TOTUM-854 human metabolites on redox status in human umbilical vascular endothelial cells (HUVECs). DHE staining: (**A**) quantification expressed as the mean red pixel value; (**B**) inverted fluorescence microscopy; (**C**) Nox2 mRNA relative expression; (**D**) Nitric Oxide release in cell culture media. Control groups (Ctrl) consist of human naïve serum exposition with or without palmitate. TOTUM-854 groups consist of human serum containing circulating metabolites from TOTUM-854 ingestion with or without palmitate. Measurements were realized in triplicate for each volunteer (n = 10 volunteers). Boxes indicate median and interquartile range (lower and upper), while whiskers indicate minimum and maximum *: *p* < 0.05; **: *p* < 0.01; ****: *p* < 0.0001; ns: *p* > 0.05.

### 3.6. Inhibition of the Inflammatory Response

Hypertensive and lipotoxic context fuels inflammation by stimulating pro-inflammatory cytokine release and promoting the recruitment of immune cells. We investigated IL-1b production and release under different conditions. As for previous parameters, metabolites alone had no impact on IL-1b release. Consistently, the induction of lipotoxic stress by palmitate doubled IL-1b production (+129%, *p* = 0.0524), while pre-incubation with TOTUM-854 metabolites potently abolished it, returning IL-1b release values close to the baseline values (−41% compared to palmitate, *p* < 0.001) (Figure 7A). We also examined MCP-1 (Figure 7B) and VCAM-1 (Figure 7C) expression, two markers known for their involvement in immune cell recruitment. Their expression profiles nicely paralleled IL-1b release. Palmitate dramatically induced the expression of both MCP-1 (*p* = 0.0506) and VCAM-1. In contrast, even if the decrease in the expression of MCP-1 and VCAM-1 by TOTUM-854 metabolites alone did not reach significance, in the presence of palmitate, TOTUM-854 metabolites successfully blunted the increased expression of both markers (−86% *p* < 0.001 and −70% *p* < 0.001, respectively).

**Figure 7 nutrients-17-01331-f007:**
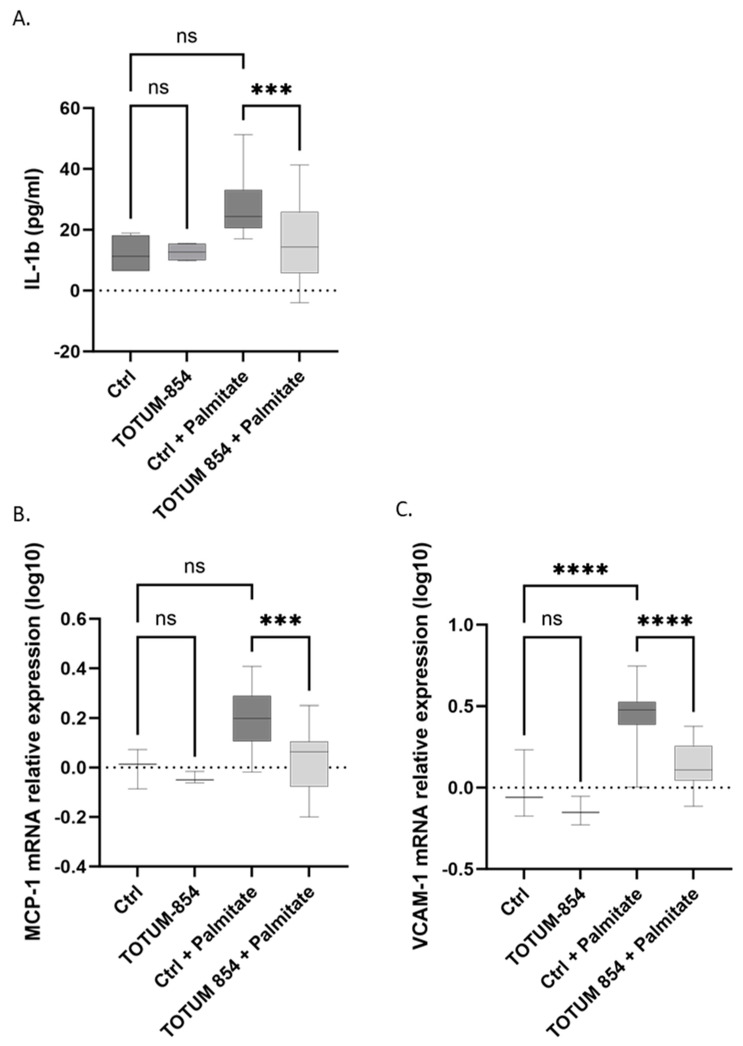
Influence of TOTUM-854 human metabolites on inflammatory mediators and cell adhesion in human umbilical vascular endothelial cells (HUVECs). (**A**) IL-1β release (pg/mL). (**B**) MCP-1 mRNA relative expression. (**C**) VCAM-1 mRNA relative expression. Control groups (Ctrl) consist of human naïve serum exposition with or without palmitate. TOTUM-854 groups consist of human serum containing circulating metabolites from TOTUM-854 ingestion with or without palmitate. Measurements were realized in triplicate for each volunteer (n = 10 volunteers). Boxes indicate median and interquartile range (lower and upper), while whiskers indicate minimum and maximum ***: *p* < 0.001; ****: *p* < 0.0001; ns: *p* > 0.05.

## 4. Discussion

In this manuscript, we first demonstrated that TOTUM-854, a plant extract blend, was able to inhibit ACE activity in vitro. Its potential health benefit was then further explored and validated in vivo as a candidate for regulating blood pressure in a mouse model of hypertension. Upon those encouraging preliminary data, we investigated the nutraceutical potential of TOTUM-854 in humans. Using an ex vivo clinical approach, we characterized the bioavailability of the blend and demonstrated that the circulating metabolites resulting from TOTUM-854 ingestion in humans were able to maintain endothelial cell function in a lipotoxic environment by (1) inhibiting ACE-1 activity, (2) limiting oxidative stress, (3) preventing the release of IL-1β, and (4) reducing the expression of MCP-1 and VCAM-1, which are both involved in inflammation and the recruitment of immune cells that drive endothelial dysfunction onset.

The relevance of each model used in this work may be questioned, including the nutritional dimensions of the dose used for the clinical investigation. Each volunteer was exposed once per clinical phase to a total of 3170 mg of TOTUM-854. If we take into consideration the amount of oleuropein, the most abundant polyphenol in the blend, the exposure dose was about 334 mg, which is consistent with other clinical studies. In Lockyer’s 2017 study, the dose of oleuropein given daily to volunteers for 6 weeks was 136 mg/d [30]. Overall, the total quantity of polyphenols provided by 3170 mg of TOTUM-854 is estimated to be around 560 mg. Since the recommended daily dose of polyphenols is about 1 g [21,31], the dose used in this study was relevant from a nutritional point of view.

In this context, about nine circulating human metabolites were detected following TOTUM-854 ingestion in humans. Among them, two were oleuropein glucuronides isomers, three were luteolin glucuronides isomers, one was a hydroxytyrosol sulfate isomer, one was a ferulic acid sulfate isomer, one was a homovanillic acid sulfate isomer, and one was a dihydrocaffeic acid sulfate isomer.

As expected, these metabolites mainly originate from olives and artichokes, which are the most abundant extracts of the blend. Olive and artichoke extracts were previously and consistently reported for their positive influence on both lipid metabolism and blood pressure. In the early 2000s, a clinical study on hyperlipidemic patients showed that daily supplementation with artichoke juice for six weeks improves arterial compliance, with increased brachial flow-mediated vasodilation [32]. More recently, in a double-blind randomized study, an olive leaf extract, characterized by its oleuropein and hydroxytyrosol content, was given to hypertensive patients for 6 weeks and led to a reduction in blood pressure and inflammatory status and improved the lipid profile [30]. One homovanillic acid sulfate isomer was detected in the serum. Interestingly, this tyrosol-derived metabolite has been known for decades to be a catecholamine metabolite involved in the reduction in blood pressure in hypertensive patients [33]. Altogether, these data on bioavailability are quite consistent with the literature and with both our preclinical and clinical observations, providing strong support for the potential health benefits of TOTUM-854 in hypertension.

Conversely, and despite their interest previously described in treating hypertension, we did not detect either garlic-, bilberry-, or piperin-derived metabolites. In an intervention study conducted over 12 weeks in hypercholesterolemic subjects, the daily supplementation of bilberry extract (320 mg/day, n = 75) led to a significant improvement in arterial compliance [34]. The same conclusions were obtained in a more recent study that was twice as long (6 months) in subjects with metabolic syndrome [35]. Piperin, administered by gavage (20 mg/kg/day) for 3 weeks, limited the increase in blood pressure induced by L-NAME in rats [36]. Additionally, two recent clinical studies reported the benefit of garlic extract on endothelial function and subsequent blood pressure in obese [37] and type-2 diabetic patients [38]. In our study, volunteers were exposed once to 3 mg of black pepper, which is lower than previous studies on the biological activity of long-term exposure to piperin (7.5 mg/day to 16 mg/day for 4 to 8 weeks) [39,40]. In fact, in this blend, piperin was used for its ability to improve the bioavailability and bioefficacy of the product with which it is associated and thus optimize the potential synergistic effects [41,42]. Regarding garlic, alliin is one of its most interesting bioactive but only represents 1% of garlic components [43]. In TOTUM-854, the quantity of alliin found in the raw blend was about 2000 times lower than oleuropein, which may contribute to explaining why its circulating metabolites remained under the detection threshold.

One of the main results of this study is the ability of TOTUM-854 human metabolites to inhibit ACE-1 activity both in vitro and ex vivo. As previously mentioned, despite several side effects, ACE inhibitors have been successfully used for decades to treat hypertension [4,13]. In this light, a few preclinical studies have shown that the efficacy of nutritional strategies in the management of hypertensive conditions may involve ACE inhibition. In a model of spontaneous hypertension in rats, a diet enriched with 3% freeze-dried blueberries for 4 or 8 weeks improved systolic blood pressure [44]. This protection involved the inhibition of the angiotensin-converting enzyme [45]. In a rat L-NAME-induced model of hypertension, garlic consumption completely prevented the increase in systolic blood pressure [17]. The hypotensive action of garlic was partly proven to rely on its ability to inhibit the angiotensin-converting enzyme [46,47,48]. Regarding isolated polyphenols, both chlorogenic acid and caffeic acids were reported to inhibit angiotensin-converting enzyme activity in a preclinical model of cyclosporine-induced hypertension [20,21]. Bilberries and garlic extracts are part of the blend, and TOTUM-854 also contains a wide range of polyphenols including chlorogenic and caffeic acids. Even though we could not detect all the specific metabolites of the different extracts, in line with the literature data, the bulk of the human metabolites derived from TOTUM-854 ingestion decreased ACE-1 activity in human endothelial cells in a lipotoxic context, likely contributing to explaining the observed benefit on endothelial cell function.

Among the molecular mechanisms that drive hypertension onset, oxidative stress and inflammation are major triggers. Endothelial inflammation stimulates pro-inflammatory cytokine production and monocyte recruitment and adhesion. These events promote the initiation of atherosclerosis, leading to hypertension development [49]. IL-1β (a pro-inflammatory cytokine), MCP-1 (a chemo-attractant cytokine), and VCAM-1 (an adhesion molecule) have been particularly highlighted in the context of endothelial cell dysfunction and hypertension [49,50]. Here, we demonstrated that human metabolites from TOTUM-854 consistently and potently reduced the release of IL-1β and the expression of both MCP-1 and VCAM-1 in a lipotoxic environment. Interestingly, in a recent randomized, double-blind, placebo-controlled study on patients with high cardiovascular risk, daily supplementation with 1200 mg of garlic extract for 4 weeks led to a potent reduction in both vascular inflammation and oxidative stress [38]. The same observations were made when the dose was reduced to 400 mg/d for a longer period of time (3 months) in obese patients [37]. Consistent with this, it has recently been reported that the hypotensive properties of olive leaf extract occur via the suppression of oxidative stress [51]. In our study, TOTUM-854 metabolites dramatically reduced the generation of ROS in HUVECs in a lipotoxic environment, as shown by DHE staining. This inhibition of DHE fully matches the observed inhibition of NOX2 expression and its role in hypertension establishment. A clinical study focused on the role of dark chocolate polyphenols on endothelial function in non-alcoholic steatohepatitis patients demonstrated a correlation between the reduction in NOX2 and the improvement of flow-mediated dilatation in the brachial artery (an index of endothelium-dependent vasodilation) [52]. The same trend was observed in children and adults with obstructive sleep apnea. NOX2 expression was associated with oxidative stress-mediated arterial dysfunction, and the decreased expression of NOX2 was inversely correlated with the improvement in FMD [53,54].

Hence, these data further support the health benefits of TOTUM-854 supplementation in the management of moderate hypertension and highlight how TOTUM-854 metabolites contribute to maintaining normal endothelial cell function in a stress context.

This study provides important initial insights and promising avenues for understanding the mechanisms of action of TOTUM-854. By integrating a clinical study with preliminary ex vivo experiments on a cellular model and an in vivo proof-of-concept in a rodent model, we offer the first characterization of TOTUM-854’s potential effects. However, additional complementary experiments are needed to fully dissect its mechanisms of action. Future studies should explore alternative in vitro models, particularly those with pre-existing NO production impairment, to further assess the potential direct or indirect effects of TOTUM-854 on NO synthesis. Moreover, investigations into additional in vivo models of hypertension will be essential to confirm and extend our findings, as well as to better understand the broader physiological relevance of TOTUM-854’s effects on vascular function. Ultimately, clinical efficacy studies in humans, specifically in populations with elevated blood pressure, will be necessary to fully establish the therapeutic potential of TOTUM-854.

## 5. Conclusions

In this study, we showed that TOTUM-854 was able to protect against hypertension in vivo and further demonstrated how such benefits could occur ex vivo at a clinical level. The bio-available human metabolites from TOTUM-854 protected HUVECs from induced lipotoxic stress. HUVEC protection was characterized by (1) decreased ACE-1 activity, (2) decreased oxidative stress, (3) the inhibition of the inflammatory response, and (4) decreased marker expression of endothelial reactivity. Using both a relevant in vivo model and a pioneering clinical ex vivo approach, altogether these data support hypertension prevention and enlighten the bio-availability of TOTUM-854 human metabolites and their role in endothelial cell protection.

Two phase II/III randomized clinical trials (NCT05469503 and NCT05370625) are ongoing to fulfill the clinical evidence of the health benefits of TOTUM-854 in maintaining endothelial cell function and managing moderate hypertension.

## 6. Patents

The human ex vivo methodology used in this study has been registered as a written invention disclosure by the French National Institute for Agronomic, Food, and Environment Research (INRAE) (DIRV#18-0058). Clinic’n’Cell^®^ has been registered as a trademark [11,22,23,24,25,26,27,28].

## Data Availability

The data presented in this study are available on request from the corresponding author. The data are not publicly available due to ethical restrictions.

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
