# Peer review of "TOTUM-854 Human Circulating Bioactives Preserve Endothelial Cell Function"

_nutrients, 2025, doi:10.3390/nu17081331_

Round 1
Reviewer 1 Report
Comments and Suggestions for Authors
The indication of statistical analysis is unusual.
Additionally, in figure 4 authors indicated that the differences between Ctrl and Ctrl + palmitate are significant. This is questionable, and surely does not indicate any relevant effect on cell viability.
The chromatograms should be included in the manuscript, also as supplementary material.
The reference section has to be formatted according to journal guidelines.
Author Response
The indication of statistical analysis is unusual.
We first thank you for your time and consideration.
We did not understand whether your comment concerned the figures or the section 2.17. Therefore, we both, brought corrections in the text and changed some figures to improve clarity.
Additionally, in figure 4 authors indicated that the differences between Ctrl and Ctrl + palmitate are significant. This is questionable, and surely does not indicate any relevant effect on cell viability.
Thanks for your remark. We understand your point regarding a relevant effect of palmitate on cell viability. We agree that, although significant, the effect of palmitate remains weak and do not enlighten a massive impact of cell viability. However, it was interesting to notice that cell growth was slightly impaired by palmitate alone while such an effect (even weak) was absent in the presence of TOTUM-854 metabolites. With the figure 4, our main objective was to show that the incubation with human serum did not alter primary cell viability/growth and that it would not represent a bias in our conclusions. Besides, since, all experiments were conducted on a 24h period with palmitate, we removed the figure 4A (48h) that may be confusing. In the light of your comment, we changed the text as following:
To avoid any bias in our conclusions, we first insured that human serum (either naïve or enriched) had no detrimental effect on HUVECs growth and viability (Figure 4). HUVECs were pre-incubated for 24 hours in the presence of either naive serum or TO-TUM-854 enriched serum, 24h-prior lipotoxic stress induction by palmitate (200μM). Data showed no difference between the different serum conditions. Human serum either naïve or enriched did not modified cell viability compared to each other or compared the control FCS treatment. In contrast, the presence of palmitate slightly but significantly reduced cell growth. Interestingly, palmitate had no effect on cells when HUVECs were pre-incubated with the circulating metabolites from TOTUM-854.
The chromatograms should be included in the manuscript, also as supplementary material.
We agree. The chromatograms were added as a supplemental figure S1.
The figure is referenced in the text as following: The chromatograms of the detected metabolites in serum are presented in the supplemental data section (Figure S1).
The reference section has to be formatted according to journal guidelines.
Thanks for your comment. According to Nutrients’ Instructions for authors, references may be in any style even if they recommend to use “style 9” if available online. We did not find this style online, however, in order to address your comment, we edited an endnote style to make it look like throughout the manuscript.
Reviewer 2 Report
Comments and Suggestions for Authors
In this manuscript (ID# nutrients-3476813), entitled “TOTUM-854 human circulating bioactives preserve endothelial 2 cells function”, authors Wauquier et al studied the effect of TOTUM-854 on vascular endothelial function using in vitro and in vivo approaches. Their results have demonstrated that TOTUM-854 reduced blood pressure in L-NAME-induced hypertension mice. In Human umbilical vascular endothelial cells, TOTUM-854 decreased palmitate-induced ROS generation, IL-1B production, and MCP-1/VCAM-1 expression. They conclude that TOTUM-854 has protective effects on endothelial function. However, the experimental design is not rigorous. The results are not reliable. Several major concerns are listed in the following paragraphs:
- In mice, TOTUM-854 attenuated hypertension induced by L-NAME, a NOS inhibitor, suggesting the antihypertensive effect of TOTUM-854 is nitric oxide dependent. However, in Fig 6D, TOTUM-854 has no effect on nitric oxide production. Please explain the controversy among those results.
- Regarding ethics, please provide animal IACUC approval number and the specific committee name, instead of “local research committee”.
- The major focus of this study is antihypertension of TOTUM-854, why healthy individuals were used instead of patients with hypertension?
- The human umbilical vein endothelial cell (HUVEC) was used to test the effect of TOTUM-854 on endothelial dysfunction in hypertension. However, umbilical vein is not directly related with regulation of blood pressure. Why HUVEC was selected for in vitro protocol, because so many primary human endothelial cell cultures are available now.
- The effect of TOTUM-854 on cell viability was examined in HUVECs (Fig 4). However, the endothelial cells are not exposed to TOTUM-854 directly in individuals after oral administration and metabolism in gastrointestinal and hepatic systems. The effect of TOTUM-854 metabolites on cell viability should be examined.
- In Fig 6, the DHE was used to measure redox status in cells after treatment with TOTUM854 and/or Palmitate. However, the DHE is usually used to detect superoxide and the half-life of super oxide is only several seconds. How this experiment was performed?
Author Response
In this manuscript (ID# nutrients-3476813), entitled “TOTUM-854 human circulating bioactives preserve endothelial cells function”, authors Wauquier et al studied the effect of TOTUM-854 on vascular endothelial function using in vitro and in vivo approaches. Their results have demonstrated that TOTUM-854 reduced blood pressure in L-NAME-induced hypertension mice. In Human umbilical vascular endothelial cells, TOTUM-854 decreased palmitate-induced ROS generation, IL-1B production, and MCP-1/VCAM-1 expression. They conclude that TOTUM-854 has protective effects on endothelial function. However, the experimental design is not rigorous. The results are not reliable. Several major concerns are listed in the following paragraphs:
In mice, TOTUM-854 attenuated hypertension induced by L-NAME, a NOS inhibitor, suggesting the antihypertensive effect of TOTUM-854 is nitric oxide dependent. However, in Fig 6D, TOTUM-854 has no effect on nitric oxide production. Please explain the controversy among those results.
Thank you for your insightful comments regarding the apparent controversy between the in vivo antihypertensive effect of T854 in L-NAME-treated mice and its lack of direct effect on NO production in HUVECs in vitro. We would like to clarify the rationale behind these observations and how they align with our study objectives.
HUVECs are not an optimal model to evaluate T854’s potential NO-related effects. In our in vitro model, HUVECs exhibit normal NO production at baseline, whereas in L-NAME-treated mice, NO production is significantly impaired. It is plausible that T854 only exerts an effect when NO production is already compromised. Nevertheless, we measured NO production in HUVECs to determine whether T854 had a direct effect on this parameter. The absence of an observed effect under these conditions provides an initial level of insight, suggesting that T854 does not directly stimulate NO synthesis in this model. However, this does not rule out the possibility that an effect might be observed in an alternative in vitro system where NO production is already impaired. Likewise, it remains possible that T854 impacts NO-related hypertension indirectly in a more complex in vivo environment.
The in vivo model was used as a proof of concept rather than a mechanistic investigation. Our primary aim in using the L-NAME-induced hypertension model was to assess whether T854 has antihypertensive potential in vivo, rather than to determine its precise mechanism of action. The L-NAME model is widely used in the evaluation of antihypertensive drugs, including those that do not act directly on NO synthesis. T854’s effect on blood pressure in L-NAME-treated mice does not necessarily imply a direct action on NO production. Many antihypertensive agents (e.g., ACE inhibitors, ARBs) are effective in L-NAME models despite not directly stimulating NO synthesis. Instead, T854 could be acting through an alternative pathway that indirectly influences vascular tone, such as ACE inhibition and the renin-angiotensin system (RAS). Moreover, ACE inhibitors are known to modulate NO production indirectly. As demonstrated in our study, T854 inhibits ACE1 activity in vitro. ACE inhibitors increase bradykinin levels, which in turn stimulate NO production via activation of endothelial NO synthase (eNOS). However, in vitro, HUVECs do not produce bradykinin, and the kallikrein-kinin system is not fully functional. This could also explain why potential T854’s indirect effect on NO may not be observable in an isolated HUVEC model but can still contribute to blood pressure regulation in vivo.
In conclusion, the findings in our study are consistent with the hypothesis that T854 acts primarily through ACE inhibition and potential other mechanisms and not via direct NO stimulation. The L-NAME-induced hypertension model provides a proof of concept for T854’s in vivo efficacy but does not define its precise mechanism of action. The lack of NO modulation in HUVECs does not contradict its antihypertensive effect, as T854’s influence on NO could be indirect and dependent on physiological conditions absent in an isolated endothelial cell model.
We hope this clarification addresses the concerns raised, and we also propose to add at the end of the discussion a section related to study limitations and future directions:
“This study provides important initial insights and promising avenues for understanding the mechanisms of action of TOTUM-854. By integrating a clinical study with preliminary ex vivo experiments on a cellular model and an in vivo proof-of-concept in a rodent model, we offer a first characterization of T854’s potential effects. However, additional complementary experiments are needed to fully dissect its mechanism of action. Future studies should explore alternative in vitro models, particularly those with pre-existing NO production impairment, to further assess potential direct or indirect effects of T854 on NO synthesis. Moreover, investigations in additional in vivo models of hypertension will be essential to confirm and extend our findings, as well as to better understand the broader physiological relevance of T854’s effects on vascular function. Ultimately, clinical efficacy studies in humans, specifically in populations with elevated blood pressure, will be necessary to fully establish the therapeutic potential of T854.”
Regarding ethics, please provide animal IACUC approval number and the specific committee name, instead of “local research committee”.
All the animal procedures were approved by the local ethics committee (C.E. Languedoc-Roussillon, France, under the number CEEA-00322.03). The text (section 2.2) was modified to include this statement.
The major focus of this study is antihypertension of TOTUM-854, why healthy individuals were used instead of patients with hypertension?
Thanks for pointing this out. In our clinical ex vivo approach, volunteers should be seen as metabolites producers rather than patients expecting health benefits with measurable clinical scores. In fact, during this acute exposition, we “just” want these volunteers to produce circulating metabolites in a standardized way. In this context, we only recruit healthy volunteers with no treatment or pathology, aged from 18 to 35, BMI ranged from 20 to 28, normal kidney and liver functions. Then, the collected serum, either naïve or enriched with circulating metabolites of interest, are used/incubated on primary cell cultures to investigate a wide range of cell activities/markers that represent the final readout/score.
The human umbilical vein endothelial cell (HUVEC) was used to test the effect of TOTUM-854 on endothelial dysfunction in hypertension. However, umbilical vein is not directly related with regulation of blood pressure. Why HUVEC was selected for in vitro protocol, because so many primary human endothelial cell cultures are available now.
You’re right endothelial cells can now be obtained from aortic, coronary, pulmonary or dermal vessels. The point is that we got previous expertise on that model so we did not want to jeopardize our protocol with another model and started with HUVECs. Besides, HUVECs is an acknowledged model regarding lipotoxic and palmitate-induced stress investigations. Experiments on aortic cells were also planned in a second step but we run out of serum. I agree that it would be relevant to repeat these investigations on different models to fulfill our study.
The effect of TOTUM-854 on cell viability was examined in HUVECs (Fig 4). However, the endothelial cells are not exposed to TOTUM-854 directly in individuals after oral administration and metabolism in gastrointestinal and hepatic systems. The effect of TOTUM-854 metabolites on cell viability should be examined.
I think there’s a misunderstanding. In figure 4 cells were not directly exposed to TOTUM-854 but to serum either naïve or containing circulating metabolites resulting from a completed a physiological human digestion of TOTUM-854. In this light, we therefore examined the effect of TOTUM-854 metabolites on cell viability.
To improve flow and clarity we modified the text (section 3.3) as following: To avoid any bias in our conclusions, we first insured that human serum (either naïve or containing circulating metabolites resulting from TOTUM-854 ingestion) had no detrimental effect on HUVECs growth and viability (Figure 4).
In Fig 6, the DHE was used to measure redox status in cells after treatment with TOTUM854 and/or Palmitate. However, the DHE is usually used to detect superoxide and the half-life of super oxide is only several seconds. How this experiment was performed?
Thanks for this relevant remark. We added details on the technical conditions used to perform the assay. The text (section 2.13) was modified as following: DHE (Dihydroethidium) Assay Kit - (ab236206, Abcam, Paris, France) was used to measure reactive oxygen species (ROS) directly in live cells. This kit uses DHE as a fluorescent probe for the detection of ROS generation notably superoxide, hydrogen peroxide but also non-specific oxidation by other sources of ROS to form ethidium (excitation 480-520 nm and emission 570-600 nm) (doi: 10.3390/antiox12122105). At the end of the incubation period with palmitate in the presence of naïve or enriched serum, cells were incubated with 5µM of DHE probe for 30 minutes at 37°C and washed prior to observation. Live cells were imaged form a ZEISS optical microscope with a B/W cooled camera and analyzed using imageJ software (NIH). Measurements were performed for each sample condition of the ten volunteers.
To further discuss your point, you’re right, the redox-sensitive fluorescent probe dihydroethidium (DHE) was first associated with superoxide anion detection which half-life is about seconds depending on the pH and temperature. However, the specificity of DHE has been debated and it is now acknowledged to detected hydrogen peroxide and non-specific oxidation by other sources of ROS to form ethidium. We added in the text the following reference in support (doi: 10.3390/antiox12122105). Therefore, in this experiment we did not really focus on superoxide but we rather used this probe as a general ROS detection tool.
Of interest, we also performed DCFDA experiments that showed exactly the same trends but due to high technical variation, the results did not reach significance and we decided not to show them.
Round 2
Reviewer 2 Report
Comments and Suggestions for Authors
The revised manuscript has been improved and no further recommendation.